# Effect of 12-Week BMI-Based Vitamin D_3_ Supplementation in Parkinson’s Disease with Deep Brain Stimulation on Physical Performance, Inflammation, and Vitamin D Metabolites

**DOI:** 10.3390/ijms241210200

**Published:** 2023-06-15

**Authors:** Zofia Kinga Bytowska, Daria Korewo-Labelle, Paweł Berezka, Konrad Kowalski, Katarzyna Przewłócka, Witold Libionka, Wojciech Kloc, Jan Jacek Kaczor

**Affiliations:** 1Division of Bioenergetics and Physiology of Exercise, Faculty of Health Sciences with Institute of Maritime and Tropical Medicine, Medical University of Gdansk, 80-211 Gdansk, Poland; zofia.bytowska@gumed.edu.pl (Z.K.B.); k.przewlocka@gumed.edu.pl (K.P.); 2Department of Physiology, Faculty of Medicine, Medical University of Gdansk, 80-210 Gdansk, Poland; daria.korewo@phdstud.ug.edu.pl; 3Department of Animal and Human Physiology, Faculty of Biology, University of Gdansk, 80-309 Gdansk, Poland; pawel.berezka@phdstud.ug.edu.pl; 4Masdiag-Diagnostic Mass Spectrometry Laboratory, Stefana Żeromskiego 33, 01-882 Warsaw, Poland; konrad.kowalski@masdiag.pl; 5Department of Neurosurgery, University Clinical Centre in Gdansk, 80-952 Gdansk, Poland; wlibionka@gmail.com; 6Department of Neurosurgery, Copernicus Medical Center, 80-803 Gdansk, Poland; wojciech.kloc@uwm.edu.pl; 7Department of Psychology and Sociology of Health and Public Health, University of Warmia and Mazury in Olsztyn, 10-719 Olsztyn, Poland

**Keywords:** Parkinson’s disease, vitamin D, deep brain stimulation, inflammation, functional tests

## Abstract

Parkinson’s disease (PD) is the second most common neurodegenerative disease. To manage motor symptoms not controlled adequately with medication, deep brain stimulation (DBS) is used. PD patients often manifest vitamin D deficiency, which may be connected with a higher risk of falls. We administered a 12-week vitamin D_3_ supplementation based on BMI (with higher doses given to patients with higher BMI) to investigate its effects on physical performance and inflammation status in PD patients with DBS. Patients were randomly divided into two groups: treated with vitamin D_3_ (VitD, *n* = 13), and supplemented with vegetable oil as the placebo group (PL, *n* = 16). Patients underwent functional tests to assess their physical performance three times during this study. The serum 25(OH)D_3_ concentration increased to the recommended level of 30 ng/mL in the VitD group, and a significant elevation in vitamin D metabolites in this group was found. We observed significant improvement in the Up and Go and the 6 MWT in the VitD group. In inflammation status, we noticed a trend toward a decrease in the VitD group. To conclude, achieving the optimal serum 25(OH)D_3_ concentration is associated with better functional test performance and consequently may have a positive impact on reducing falling risk in PD.

## 1. Introduction

Parkinson’s disease (PD) is one of the most common neurodegenerative diseases worldwide. It affects approximately 1–2% of people over 60 years of age, and the number grows with age. In the future, the number of patients with PD is expected to be even higher because of the aging society [1]. Currently, PD is incurable, and the main aim is to alleviate symptoms, ease suffering, and slow down the development of the disease in order to help the patients to cope better with their daily living activities. Symptoms may be divided into motor and non-motor. The main motor symptoms are bradykinesia, postural instability, and resting tremors. Non-motor symptoms include depression, dementia, sleep disorders, and subtle personality changes. In PD, dopamine production is impaired because of degeneration of the substantia nigra [2,3,4,5,6]. Precursors of dopamine are the most commonly used pharmaceuticals, and L-dihydroxyphenylalanine (L-dopa) is the most widely used [7]. When pharmacological treatment ceases to control symptoms adequately, deep brain stimulation (DBS) is used to mitigate motor symptoms and decrease drug dosage. In PD, electrodes are usually implanted in the subthalamic nucleus [8,9,10].

Vitamin D is one of the most significant vitamins in our organism. Calcium and phosphate homeostasis are regulated by vitamin D and parathormone [11]. Vitamin D may be synthesized through the skin due to UV-B radiation. Either can be provided with a diet, but it is difficult to achieve acceptable results; thus, dietary supplements of vitamin D3 are usually recommended. After sun exposure, the skin converts 7-dehydrocholesterol to cholecalciferol. It is then hydroxylated to 25-hydroxycholecalciferol (25(OH)D_3_) in the liver. This metabolite is a non-active form of vitamin D, even though it is used to determine if the deficiency occurs due to longer maintenance in the blood when compared to its active form. It is activated by an enzyme (1α-hydroxylase) to 1,25-dihydroxycholecalciferol (1,25(OH)_2_D_3_), which is an active form. On our latitude, a very common problem is the deficiency of vitamin D due to the low insolation during the year [12,13,14]. In PD patients, the serum concentrations of 25(OH)D_3_ are lower than in age-matched healthy controls; therefore, supplementation seems to be relevant [15]. Vitamin D deficiency may lead to several disturbances in humans. It can enhance the risk of depression, increase the levels of inflammatory markers, influence muscle weakness, and generate a higher risk of falls in PD patients [13,16].

In patients with PD, inflammation is often increased when compared to healthy age-matched controls [17]. In the study from 2020, researchers demonstrated a correlation between high-sensitivity C-reactive protein (hs-CRP) concentrations and the likelihood of developing PD. Specifically, the higher the levels of hs-CRP, the greater the risk of PD [18]. On the other hand, in the study conducted in 2018, the serum concentration of hs-CRP was lower in the PD group than in the control group [19].

Physical activity is the healthiest and easiest way to improve the quality of daily living in PD patients. Many studies have proven that exercises such as aerobic exercise, Nordic walking, resistance/strength training, choreotherapy, or Tai Chi successfully help slow down the development of PD, but the key is regularity. Physical activity might be low-cost and additional action to support the pharmacotherapy [20].

Therefore, vitamin D_3_ supplementation may have beneficial effects on disease progression. Vitamin D deficiency also leads to elevated levels of inflammation. We assume that increasing the serum concentration of 25(OH)D_3_ to the optimum level from the deficiency can reduce inflammation, and improving muscle condition may lower the risk of falls.

Supplementation of vitamin D_3_ is still not as common as it should be when talking about PD patients [21]. As mentioned above, patients with PD have a lower serum 25(OH)D_3_ concentration than age-matched healthy controls. Therefore, the doses of vitamin D_3_ should be significantly higher than those in healthy people. We suggest that the dosage should be based on the patient’s BMI—the higher BMI, the higher percentage of body fat in an organism in this group of patients. We have already checked this relationship in our preliminary study. It is known that a higher fat content in the organism may influence vitamin D metabolism and reduce its beneficial actions; higher doses should be applied in people with overweight or obesity [22].

There are a small number of studies on the application of vitamin D_3_ supplementation in PD, particularly when considering PD patients with DBS. If there are any, the dosage of vitamin D_3_ is standardized for healthy people—the disease or BMI value is not included in the dose determination [23]. Our research is designed in a way that may help to check how BMI-based supplementation can affect PD.

Therefore, the aim of this study was to explore how a 12-week BMI-based vitamin D_3_ supplementation would affect the results of functional tests and serum concentration of vitamin D metabolites and hs-CRP in serum in PD patients treated with DBS.

## 2. Results

### 2.1. Demographic Characteristics

Fifty patients were enrolled in this study; however, only twenty-nine completed the intervention. Three of the enrolled patients were excluded because they did not meet the inclusion criteria, and five of them resigned due to the COVID pandemic. During the follow-up, we lost two patients from the VitD group and five patients from the PL group because they did not show up on the second visit for functional tests due to problems with transport. Six patients from the VitD group resigned from this study due to unforeseen circumstances. All patients who completed the intervention were Caucasian and met the inclusion criteria. No statistically significant differences were observed between the groups. The average age was 63 ± 9 years old in the VitD group and 66 ± 6 years in the PL group. The mean height and body mass were respectively 169 ± 12 cm and 78 ± 11 kg in the VitD group, and 174 ± 8 cm and 80 ± 20 kg in the PL group. There were six men (M) and seven women (W) in the VitD group and thirteen M and three W in the PL group. The time from DBS implantation varied from three to five years in both groups. In the off-medication condition, mean motor improvement after DBS stimulation, as assessed with UPDRS part III, was 45% (range 35–60%). The mean L-dopa-equivalent medication dosage reduction was 35%. Stimulation parameters were: monopolar stimulation in all patients, frequency 130 Hz, pulse duration 60 us, current 1.8–3.8 mA (in twenty patients with constant current stimulators), and voltage 1.9–4.2 V (in nine patients with constant voltage stimulators). Hoehn and Yahr’s (H&Y) scores in both groups were 2.5. Duration of the disease varies from 8 to 13 years in the VitD and PL groups. In the VitD group, there were three patients with normal weight, eight who were overweight, and two with obesity. In the PL group, there were six patients with normal weight, eight who were overweight, and two with obesity. The characteristics of patients are shown in Table 1.

### 2.2. Vitamin D Metabolites

The effect of supplementation on vitamin D metabolites is shown in Figure 1. We did not find any statistically significant differences between groups at the T0 in all metabolites. In both groups, at the beginning of this study, 25(OH)D_3_ deficiency was found in the serum of PD patients (<30 ng/mL). After vitamin D_3_ supplementation in the VitD group, the serum concentration of 25(OH)D_3_ reached in T2 34.99 ± 12.27 ng/mL, and it was statically significantly higher as compared to 25.55 ± 8.94 ng/mL at T0 (*p* < 0.0006; Figure 1A). There was also a statistically significant change when compared to the PL group 21.98 ± 10.91 ng/mL at T0 (*p* < 0.05; Figure 1A). We did not find any statistically significant changes in the serum concertation of 25(OH)D_2_ in both groups at T0 (0.46 ± 0.33 ng/mL in the PL group and 0.44 ± 0.18 ng/mL in the VitD group) and T2 (0.57 ± 0.45 ng/mL in the PL group and 0.46 ± 0.81 ng/mL in the VitD group) (Figure 1B). The product of catabolism of 25(OH)D_3_—24,25(OH)_2_D_3_ was elevated significantly in the VitD group at T2 (2.77 ± 1.02 ng/mL) versus T0 (2.09 ± 1.09 ng/mL) (*p* < 0.05; Figure 1C). There was also a statistically significant change when compared to the PL group of 1.67 ± 1.15 ng/mL at T0 (*p* < 0.05; Figure 1C). There was no change in the serum concentration of this metabolite in the PL group at T2 (1.32 ± 0.81 ng/mL) to T0 (1.67 ± 1.15 ng/mL). The last metabolite of vitamin D that we measured was epi-25(OH)D_3_. The concentration of epi-25(OH)D_3_ was 0.83 ± 0.54 ng/mL and 1.03 ± 0.37 ng/mL, respectively, in the PL and VitD groups at T0, and 0.79 ± 0.54 ng/mL and 1.67 ± 0.70 ng/mL, respectively, in PL and VitD group at T2. We found a significant increase in the VitD group (*p* < 0.005) and a significant change when compared with the VitD group at T2 to the PL group at T0 (*p* < 0.005; Figure 1D). We have also found a strong positive correlation between 25(OH)D_3_ and 24,25(OH)_2_D_3_ (*p* < 0.0001; Figure 2A), as well as between 25(OH)D_3_ concentration and epi-25(OH)D_3_ (*p* < 0.0001; Figure 2B).
Figure 1The effect of supplementation on vitamin D metabolites. (**A**) Changes in serum concentrations of 25(OH)D_3_. (**B**) Changes in serum concentrations of 25(OH)D_2_. (**C**) Changes in serum concentration of 24,25(OH)_2_D_3_. (**D**) Changes in serum concentrations of epi-25(OH)D_3_. * *p* < 0.0006, ** *p* < 0.005, *** *p* < 0.05. T0—before the supplementation; T2—after 12 weeks of supplementation; PL—placebo group; VitD—vitamin D_3_ group; whiskers refer to standard error (SE).
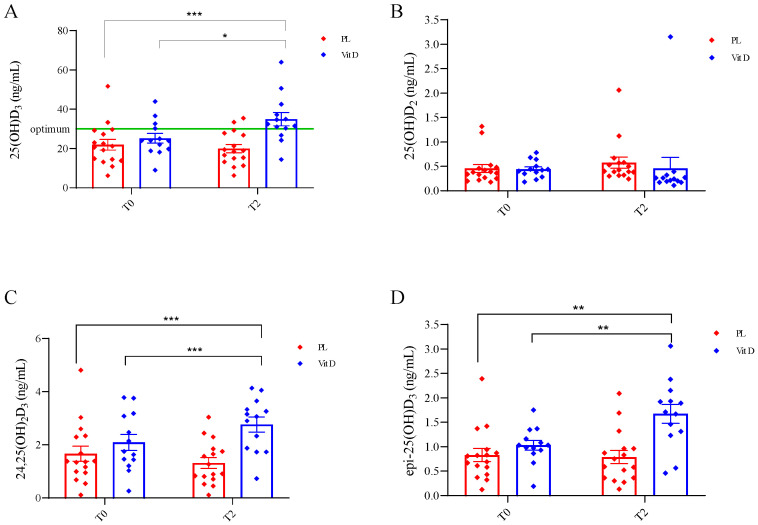



### 2.3. Functional Tests

The effect of supplementation on functional tests is presented in Figure 3. We did not find any statistically significant differences between groups at the T0 in all tests. In the TUG, we found a statistically significant change in the VitD group after the supplementation when compared with T0 (13.69 ± 5.10 s) to T1 (11.96 ± 3.44 s) (*p* < 0.05; Figure 3A), as well as when compared T0 to T2 (11.46 ± 3.80 s) (*p* < 0.005; Figure 3A). There were no changes in the PL group in this test at all three time points (T0—10.65 ± 2.44 s, T1—10.56 ± 2.73 s, T2—9.86 ± 1.63 s; Figure 3A). In the 6 MWT, we observed significant change when comparing the VitD group T0 (316.68 ± 93.45 m) to T2 (350.29 ± 96.28 m) (*p* < 0.05; Figure 3B), and no changes when considering T1 (339.99 ± 91.43 m). There were no changes in the PL group (T0—381.23 ± 74.74 m, T1—379.99 ± 56.5 m, T2—377.61 ± 75.6 m; Figure 3B). We did not notice any statistically significant changes in 10 MWT at all three time points in the PL and the VitD group, respectively (T1—9 ± 1.59 s, 10.39 ± 3.24 s; T2—8.46 ± 1.00 s, 9.88 ± 2.38 s; T3—8.66 ± 1.43 s, 9.31 ± 2.47 s; Figure 3C).

### 2.4. General Inflammation Status

The effect of supplementation on general inflammation status was measured by serum hs-CRP concentration, as shown in Figure 4. We did not observe any significant changes in serum concentration of hs-CRP in both groups; however, we noticed a trend toward a decrease in the VitD group T0 (3091.12 ± 1358.19 ng/mL) vs. T2 (2454.32 ng/mL ± 1325.18) (Figure 4). We did not observe any changes in the PL group (T0—3167.04 ng/mL ± 2103.74; T2—3510.49 ± 1852.66 ng/mL). The lack of significant alteration was probably caused by the large SD, both in and between the groups.

## 3. Discussion

To the best of our knowledge, this is one of the first studies where PD patients with DBS were supplemented with vitamin D_3_, and it is the first when the dosage depends on the patient’s BMI. Previous studies showed that patients with PD suffer from vitamin D deficiency and have a lower serum 25(OH)D_3_ concentration than healthy age-matched controls [15,24,25,26,27], and our results are in line with their outcomes. We found that after supplementation based on patients’ BMI, the serum concentration of 25(OH)D_3_ increased above the optimum level (30 ng/mL), which shows that considering the BMI when calculating the dosage is a good concept. We also measured the metabolites of vitamin D. 24,25(OH)_2_D_3_ is a product of the catabolism of 25(OH)D_3,_ and its production is regulated by 25-hydroxyvitamin D-24-hydroxylase. Recently, Couchman and co-workers reported that, usually, it is between 2% and 20% of 25(OH)D_3_ (average 10%), but when it comes to the deficiency of 25(OH)D_3,_ it might become undetectable [28]. In our study, this metabolite was significantly elevated in the VitD group after supplementation and was approximately 8.5% of 25(OH)D_3_. In the PL group, it was also detectable but varied around 5.9% of 25(OH)D_3_. Another metabolite that we measured was epi-25(OH)D_3_, which is a product of the epimerization of 25(OH)D_3_. The role of the epimerization pathway has not been sufficiently investigated, but it has been confirmed that 25(OH)D_3_ is positively correlated with epi-25(OH)D_3_ [29]. Both metabolites increased significantly after supplementation, suggesting that the metabolism of vitamin D was not disturbed by DBS treatment. Strong positive correlations were found between 25(OH)D_3_ and 24,25(OH)_2_D_3_, and between 25(OH)D_3_ and epi-25(OH)D_3_. We also detected that after the supplementation in the VitD group time needed to complete TUG was significantly shorter than at the beginning, which may be related to attenuating the risk of falling. In the 6 MWT, the distance covered by the VitD group improved after the supplementation period, and this alteration also reached significance.

DBS is one of the treatments used to treat motor symptoms in PD [8]. However, most researchers concentrate on PD patients only receiving pharmacological treatment during intervention with vitamin D_3_ supplementation [23,30]. We aimed to determine whether DBS influences the efficiency of supplementation. In the current study, before and after the intervention, none of the metabolites differed from those in PD patients without DBS in other studies. PD patients with DBS also suffer from serum 25(OH)D_3_ deficiency, and after vitamin D3 supplementation based on patients’ BMI, the concentration increased to the optimum level. After DBS was implanted in PD patients, a new, lower dose of L-dopa was established. During L-dopa intake, numerous disturbances occur in organisms. For example, an increase in serum homocysteine (Hcy) concentration may decrease muscle strength and negatively influence bone turnover and, as a result, decrease bone density, which may increase the risk of falls and fractures [20,31]. One of the main roles of vitamin D is to enhance bone density. Combined DBS and vitamin D_3_ supplementation may reduce the influence of L-dopa on bones and protect them from fracture. In a study conducted in a mouse model of Parkinson’s disease, the authors suggested that supplementation with vitamin D_3_ may decrease the required dosage of L-dopa and reduce its side effects. The mice were treated for two weeks after the induction of PD with 6-hydroxydopamine (6-OHDA) and vitamin D, L-dopa, or vitamin D + L-dopa for 21 days. Vitamin D was found to reverse the effects of 6-OHDA on dopamine metabolism, including behavioral deficits and oxidative stress. Dopamine metabolism also increases the effect of L-dopa [32]. In another study on patients with psoriasis, vitamin D supplementation reduced the concentration of Hcy after three months of administration [33]. Although we did not measure Hcy concentrations in our study, the results from other studies suggest that vitamin D supplementation may have beneficial effects on these parameters in patients with PD. Further studies are required to confirm this phenomenon. Regarding motor symptoms, a study by Habibi et al. showed that vitamin D did not help reduce dyskinesia induced by L-dopa [34].

Recently, we showed that vitamin D deficiency is associated with many disorders in the organism, including muscle atrophy [35] which may lead to an increased risk of falls. In a 2019 study, the authors indicated that PD patients with lower concentrations of vitamin D had a higher frequency of falls [26]. Moreover, Nocera et al. reported that shorter TUG test time in PD patients was associated with a lower risk of falls [36]. In our study, the supplementation time needed to complete the TUG was significantly shorter than at the beginning, which may be related to a decrease in the risk of falling. We also assessed the 6 MWT, which may be associated with the cardiovascular, respiratory, and locomotor systems in the elderly population [37], including PD patients. Moreover, the gait speed (m/s = distance (m)/360 (s)) can be calculated from the test results, and a speed less than 1 m/s is correlated with a higher risk of falls in the elderly population [38]. In one study, 2021 results of the 6 MWT improved after vitamin D_3_ supplementation in older adults [39]. Our results showed that after the supplementation, the distance reached by the patients was also significantly improved. Moreover, gait speed improved in the supplemented group, and it was also near the statistically significant difference (2.6% improvement compared to −1.0% in the placebo group). The last test we conducted with our patients was the 10 MWT, which has been frequently used in PD because of its high test-retest reliability [40]. In this test, we did not find any statistically significant differences.

Patients with PD are often non-active [41]. Therefore, we prompted them to increase their daily activities by walking a determined number of steps per day. Unfortunately, we did not have a professional device to control performance; therefore, we did not include physical activity in our results. We may assume that, when the steps are monitored, the results would be even better in functional tests when combined with vitamin D_3_ supplementation. In our review from 2021, we presented that regular physical activity is one of the most available and low-cost additions to PD therapy [20].

Dopamine loss is observed in PD due to the degeneration of the substantia nigra [4]. Studies on the effects of vitamin D on dopamine levels in patients with PD are scarce, and patients treated with DBS are virtually non-existent. Therefore, when we consider studies on animal models, we find some interesting results. In a study that we have already mentioned from 2022 in the 6-OHDA-induced PD mouse model, the authors reported that vitamin D treatment protected dopamine metabolism. Improvement in dopamine metabolism was observed in all groups, and the best results were observed in the group that received vitamin D + L-dopa [32]. In a study conducted in 2022, the authors showed that vitamin D_3_ supplementation combined with exercise in hemiparkinsonian rats reduces the effects of 6-OHDA. It was reported that vitamin D_3_ supplementation elevated dopamine concentration and attenuated oxidative stress in the brain [42]. Smith and co-workers reported that long-term treatment with vitamin D_3_ improved dopamine metabolism and increased the concentration of dopamine in the striatum of rats with PD induced by 6-OHDA [43]. Moreover, a study from 2018 showed similar results, indicating that vitamin D supplementation may reverse the effect of the 6-OHDA lesion, which is a decrease in dopamine. They also pointed out that vitamin D supplementation had anti-inflammatory effects [44]. Furthermore, in our previous study, it was shown that reaching an optimal serum concentration of 25(OH)D_3_ in patients with low back pain reduced markers of inflammation and decreased the intensity of pain [45]. In contrast, in a review from 2016, the authors showed that vitamin D did not have anti-inflammatory properties. They suggested that this ability may be more noticeable during sudden inflammation than during diseases that last a long time [46]. However, several studies have reported higher levels of inflammation markers in patients compared to healthy matched controls. [17,18,47]. In addition, Song with co-workers found that in PD patients, regardless of the onset age of the disease, the concentration of hs-CRP was higher than that in the control group [48]. A study conducted in 2019 investigated blood biomarkers that could be useful in predicting PD prognosis, revealing that higher hs-CRP and lower vitamin D concentrations are associated with worse daily living activities [49]. Moreover, in the study, in older adults, the authors showed that elevated ultra-sensitive CRP was associated with falls [50]. In the present study, we estimated hs-CRP, the general inflammation status in PD patients. After vitamin D_3_ supplementation in PD patients, we observed a trend toward a decrease in serum hs-CRP. However, it did not reach statistically significant differences. Therefore, it is substantial to note that the lack of significant changes in hs-CRP levels in PD patients could be attributed to a massive standard deviation and a small patient population or other unknown factors. On the other hand, from a biological perspective, a decrease of approximately 1000 ng/mL in PD patients after 12 weeks of vitamin D_3_ treatment compared to the PL group is noteworthy.

There are several important limitations of this study. First, the sample size is small as recruitment of PD patients with implanted DBS was challenging during the covid pandemic. Nevertheless, only 29 patients without missing data were included in the final analysis. Consequently, although the power of statistical analysis was compromised, the differences between groups proved to be sufficient to show statistical significance. Second, there is a lack of a true control group without DBS to check the impact of the stimulation on the supplementation. Third, we did not measure the complex of B vitamins, which are tightly associated with L-dopa treatment and might influence the results of functional tests. Fourth, patients did not have a dedicated device to control the number of steps per day; they used their mobile phones. Because of that, the quality of control of whether they performed the required physical activity is limited. Therefore, further studies are needed to confirm these results and broaden the investigation process.

## 4. Materials and Methods

### 4.1. Design of the Study

This study was a randomized, double-blind, placebo-controlled clinical trial (NCT04768023) with a 12-week supplementation of vitamin D_3_ in PD treated with DBS. This study was approved by the Independent Bioethics Committee for Scientific Research at the Medical University of Gdansk; the approval number is NKBBN/522-648/2019, date of approval 3 December 2019. Patients were recruited from the neurosurgery unit of the Hospital of Nicholas Copernicus. After recruitment, patients were randomly assigned to receive vitamin D_3_ (VitD) or a placebo (PL). They were divided into 1:1 ratios. The randomization was made with Excel random number generator. The identical bottles with vitamin D_3_ and placebo received numbers from 1 to 42. The randomization and product allocation were conducted by an independent researcher who was not engaged in any other study procedure. Patients and investigators were blinded to the intervention assignment during this study. All patients and personnel involved in the data analysis were blinded until the database was analyzed and the intervention assignment was relevant. The unblinding was performed after all data analyses.

### 4.2. Participants

Patients were recruited by invitation from the Hospital of Nicholas Copernicus from the Neurosurgery Unit. The recruitment was from November 2019 to February 2022. The intervention was made during the fall/winter season to avoid the sun exposure that is possible during the spring/summer season. Fifty patients were enrolled in this study (Figure 5).

The inclusion criteria were: agreement to take part in research, subthalamic nucleus deep brain stimulation (STN-DBS) treatment, lack of supplementation of vitamin D_3_ before the research, no serious comorbidity (tumor, cerebrovascular disease, cardiorespiratory compromise, forced dementia, etc.), and declaration of involvement.

### 4.3. Intervention

Supplementation lasted for 12 weeks for all groups. Supplementation of vitamin D_3_ was based on the participants’ Body Mass Index (BMI). The BMI was measured by using the TANITA scale. The doses were set based on the literature. The recommended dose for healthy young adults is 1000–2000 International Units (IU)/day [51]. For obese and overweight individuals, the doses are higher and range between 2500 and 4800 IU/day [52,53,54]. Taking these recommendations into consideration, along with the fact that serum concentration in PD patients is insufficient and lower than in healthy age-matched controls, the following doses were prescribed: for BMI under 25, 4000 International Units (IU)/day; for BMI between 25 and 30, 5000 IU/day; and for BMI over 30, 6000 IU/day. The placebo group received a matching placebo treatment. Vitamin D_3_, as well as the placebo, were in the same bottles with no labels on them. Patients were prompted to complete 3500 steps per day in the first week of the research and finish the research by completing 7500 per day.

### 4.4. Visit Program and Material Collection

There were three meetings with the patients. During the baseline visit (T0), there were blood collections, patients signed the agreement to take part in the research, then performed functional tests, and finally, received their first supplementation bottles. During the second visit after six weeks (T1), the functional tests were performed, and supplementation was replenished. On the third visit after six weeks (T2), the blood was collected for the second time, and functional tests were performed again. The visit programme is shown in Figure 6.

The blood was collected into test tubes with a clot activator, then centrifuged at 4000× *g* for 10 min at 4 °C. The obtained serum samples were aliquoted and stored at −80 °C until the analysis.

### 4.5. Functional Tests

The functional tests that we assessed were: the test Up and Go (TUG), the 6 min walk test (6 MWT), and the 10 m walk test (10 MWT). All tests were performed during the on-phase of the usual anti-PD medications. All tests were explained and demonstrated before they were performed. If the first attempt was successful, then the patient followed the next attempt. If any mistake appeared, the patient waited for a minute and tried again. During every meeting, the procedure of each test was explained to the patients. In the TUG, patients were instructed to stand up from the chair, walk 3 m, turn around, return to the chair, and sit down as fast as possible without running. The 6 MWT was administered in a 15 m line in a corridor. Patients were instructed to go back and forth, without running the 15 m line, as far as possible in 6 min. The total distance was measured after 6 min and then used for the analysis. In 10 MWT, we measured the time needed to walk through 10 m from a standing position.

### 4.6. Measurement of Vitamin D Metabolites

The vitamin D profile was estimated in serum by the mass spectrometry method. We apprised level of 25(OH)D_3_, 25(OH)D_2_, 24,25(OH)_2_D_3_, and epi-25(OH)D_3_. The isotope dilution method by liquid chromatography combined with tandem mass spectrometry technique (LC-MS/MS) was used. All samples were prepared and analyzed with the Eksigent ExionLC analytical HPLC system with a CTC PAL autosampler (Zwinger, Switzerland) combined with QTRAP^®^ 4500 MS/MS system (Sciex, Framingham, MA, USA).

### 4.7. Measurement of C-Reactive Protein

The measurement of C-reactive protein was made by using a Demeditec hsCRP ELISA Assay Kit (DE740011, DEMEDITEC Diagnostics GmbH, Kiel, Germany) according to the manufacturer’s instructions. All of the samples were analyzed in a microplate reader, Thermo Scientific Multiscan Go (Thermofisher Scientific, Vartaa, Finland). We followed the instructions.

### 4.8. Statistical Analysis

For the analysis, we used the statistic program Statistica 13, StatSoft Inc., (Tulsa, OK, USA). We took into consideration only full data from patients who completed all interventions (*n* = 29). Data were previously tested for normality using the Shapiro–Wilk W-test. The descriptive statistics for both background information and examination of the trends in the analyzed parameters with mean values of 95% confidence interval were used. The ANOVA test for repeated measures was used for the statistical analysis. Spearman rank correlation was calculated. The statistical significance was set at *p* < 0.05. To establish statistical significance, we applied an analysis of variance (ANOVA) with Tukey’s post hoc test.

## 5. Conclusions

The present findings indicate that 12 weeks of vitamin D_3_ treatment in patients with PD with implanted DBS induces changes in the concentration of vitamin D serum metabolites and improves functional tests. These changes may have beneficial effects with an attenuated risk of falls, which is a common problem in PD. It is worth emphasizing that these changes are associated with the dose of vitamin D_3_ and are dependent on the patient’s BMI (the higher the BMI, the higher the dose). Although these changes occurred, there was no difference in the serum hs-CRP concentrations between the groups. This study shows for the first time that the cumulative effect of DBS implementation and BMI-based vitamin D_3_ supplementation on selected blood markers and functional tests might have positive effects on reducing the number of falls, which seems to be essential for PD patients.

## Figures and Tables

**Figure 2 ijms-24-10200-f002:**
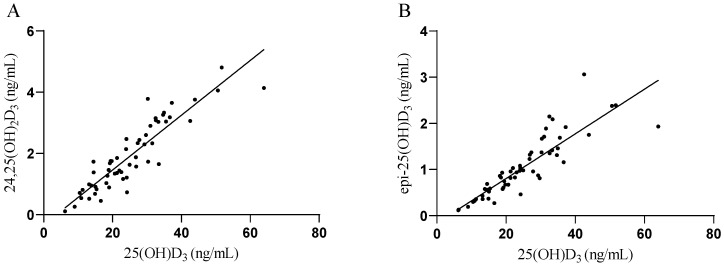
Relationships between (**A**) 25(OH)D_3_ and 24,25(OH)_2_D_3_; (**B**) 25(OH)D_3_ and epi-25(OH)D_3_. (**A**) Spearman r = 0.9, *p* < 0.0001, 95% confidence interval: 0.83 to 0.94; (**B**) Spearman r = 0.92, *p* < 0.0001, 95% confidence interval: 0.86 to 0.95.

**Figure 3 ijms-24-10200-f003:**
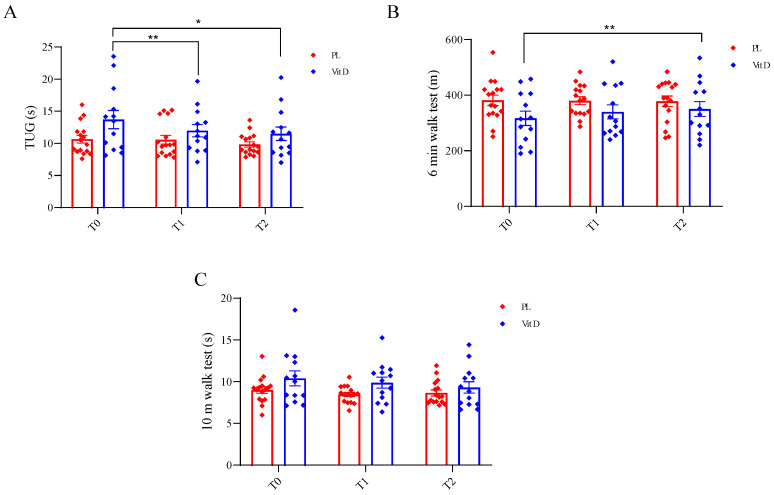
Effect of supplementation on functional tests. (**A**). Changes in time duration in TUG. (**B**). Changes in the distance in 6 min walk test. (**C**). Changes in time duration in 10 m walk test. * *p* < 0.005, ** *p* < 0.05. TUG—test Up and Go; T0—before the supplementation; T1—after 6 weeks of supplementation; T2—after 12 weeks of supplementation; PL—placebo group; VitD—vitamin D_3_ group whiskers refer to SE.

**Figure 4 ijms-24-10200-f004:**
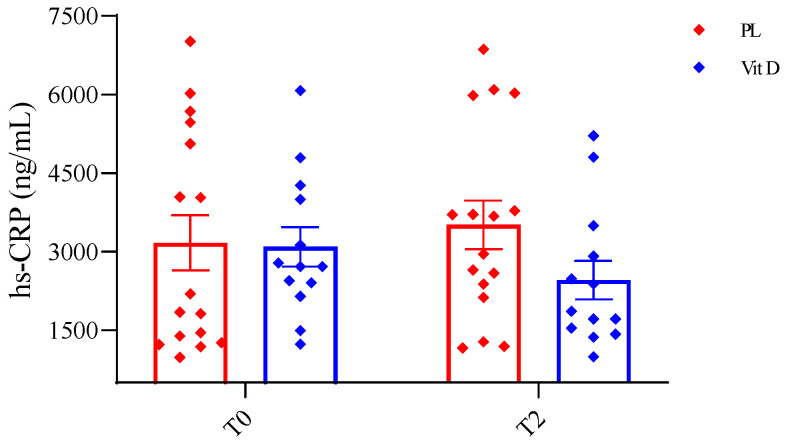
Effect of inflammation on general inflammation status. hs-CRP—high sensitivity C-reactive protein; T0—before the supplementation; T2—after 12 weeks of supplementation; PL—placebo group; VitD—vitamin D_3_ group; whiskers refer to SE.

**Figure 5 ijms-24-10200-f005:**
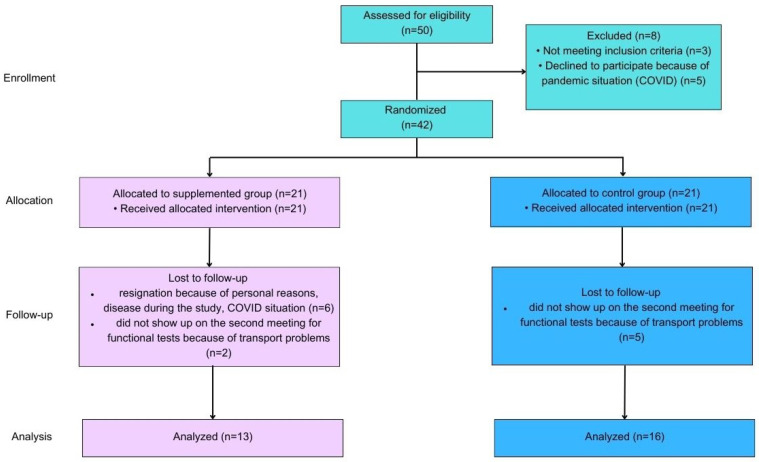
Flow diagram of this study.

**Figure 6 ijms-24-10200-f006:**
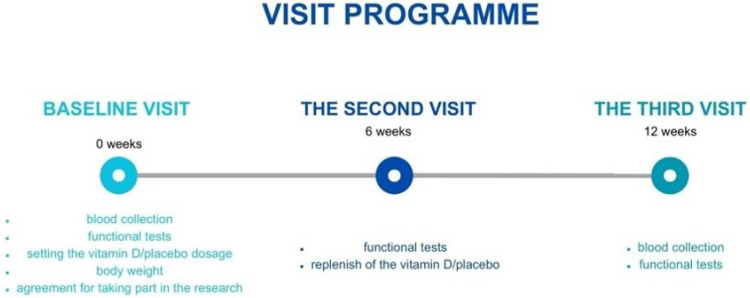
Visit programme.

**Table 1 ijms-24-10200-t001:** Patient characteristics.

	VitD Group (*n* = 13)	PL Group (*n* = 16)
Age (years)	63 ± 9	66 ± 6
Sex	6 M, 7 W	13 M, 3 W
Height (cm)	169 ± 12	174 ± 8
Body mass (kg)	78 ± 11	80 ± 20
DBS implantation	3–5 years ago	3–5 years ago
H&Y	2.5	2.5
Duration of the disease	8–13 years	8–13 years
BMI	≤25 → 3	≤25 → 6
	25–30 → 8	25–30 → 8
	≥30 → 2	≥30 → 2

## Data Availability

The data sets used and/or analyzed during the current study are available from the corresponding author upon reasonable request.

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
