# Peer review of "Effect of 12-Week BMI-Based Vitamin D3 Supplementation in Parkinson’s Disease with Deep Brain Stimulation on Physical Performance, Inflammation, and Vitamin D Metabolites"

_ijms, 2023, doi:10.3390/ijms241210200_

Round 1

Reviewer 1 Report

I am glad to be able to assess the quality of this manuscript. Please, find my commentary below.

Minor remarks:

- Line 335: '(...) willingness to work' - what did the Authors mean by using this term?

- I could not come across any guidelines mentioned in this manuscript in terms of different dosage of VitD in context of variable BMI. Could a reference be given in the manuscript? Is there some medical consensus on the dosage of VitD in terms of variable BMI?

Major remarks:

- The study itself is meticulously planned and has been registered as a clinical trial, which increases its credibility. Figures 5 amd 6 let the Readers follow the trial in a step-by-step manner. However, there are several potentially limiting factors which should be mentioned in the discussion section. Firstly, the supplementation with VitD varied depending on BMI - a factor which was higly heterogenic in the population sample (as seen in Table 1). Given that this factor affected the dosage of VitD, the population sample should be significantly greater than the one featured in this study. 'BMI-dose' should be an effect which interacts with 'time' in the repeated measured ANOVA. Moreover, as the population sample is unbalanced in terms of sex, the 'time*sex' interaction could not, likewise, be investigated. This laborous trial could have been more informative than just analyzing the time-wise variability in vitamin D3 and hsCRP concentrations. More timepoints, proper study design and thorough planning of the composition of the population sample would possibly allow drawing more interesting conclusions. As there is a lack of control group for the 'PD' effect (patients with no PD of the comparable population sample composition), it could not be answered whether the dynamics of the changes of some studied parameters (such as the concentration of VitD metabolites) is different in PD vs. no PD upon supplementation.

- The 'Statistical Analysis' section leaves as impression that the assumptions of the used methods were not thoroughly checked or accounted for (in case of their lack). RM-ANOVA assumes sphericity, which is often not found in empirical data. Therefore, corrections are performed to account for the lack of sphericity. I would advise performing the Greenhouse-Geisser or Huynh-Feldt corrections, being: more or less conservative, respectively). Since the Authors knew a priori that they wanted to check the time-wise differences, the utilization of contrast analysis would be more suitable than the Tukey's HSD post-hoc. 

- I could not see in any of the plot descriptions what the whiskers mark. Are they refering to 95% CI of the mean, SD or SE? Time-related analysis should show SD and/or SE.

- Line 389: The Pearson's r does not describe the rank correlation. It is based on residuals from the observed values vs. mean values.

With my remarks in mind, although I believe that the study could be published in an indexed journal, I would not recommend publishing it in IJMS since the study does not investigate covariates and features results based on a small-size, heterogenic population sample.

Minor spelling and grammatical checks ought to be performed.

Author Response

Reviewer#1:

Dear Reviewer,

The authors of the manuscript would like to thank you for all your comments and suggestions concerning improvements of this paper. We followed them to revise the manuscript or tried to explain any ambiguities in the point-by-point response.

I hope that all changes, which have been made would be satisfactory to Reviewer#1.

I am glad to be able to assess the quality of this manuscript. Please, find my commentary below.

Minor remarks:

- Line 335: '(...) willingness to work' - what did the Authors mean by using this term?

Thank you for your question. By this term, we wanted to say that patients that took part in our experiment declared to take part in every meeting and to take the applied dosage of the supplementation. We have changed this sentence.

“The inclusion criteria were: agreement to take part in a research, subthalamic nucleus deep brain stimulation (STN-DBS) treatment, lack of supplementation of vitamin D3 before the research, no serious comorbidity (tumor, cerebrovascular disease, cardiorespiratory compromise, forced dementia, etc.), and declaration of involvement.”

- I could not come across any guidelines mentioned in this manuscript in terms of different dosage of VitD in context of variable BMI. Could a reference be given in the manuscript? Is there some medical consensus on the dosage of VitD in terms of variable BMI?

Thank you for this question.

The usual doses for healthy, young adults of vitamin D varies between 800 – 2000 IU/day. Obesity and overweight are the factors that influence vitamin D metabolism. Adipose tissue sequestrates vitamin D and the catabolism of vitamin D is increased inside it. As a result, the doses for elderly people with obesity or overweight should be higher than for healthy, young adults. [doi: https://doi.org/10.3390/life13030808, https://doi.org/10.1007/s11154-021-09693-7, 10.23736/S2724-5985.21.02955-7, 10.1210/clinem/dgab296, https://doi.org/10.1016/j.jsbmb.2012.12.003] Moreover in PD deficiency of vitamin D is more common than in healthy individuals, thus higher doses may be necessary [doi: 10.1080/1028415X.2020.1840117]. Therefore, we assumed that the vitamin D dose would be based on the patient’s BMI. To our knowledge, this is the first study that considers BMI in determining the dose of vitamin D. The usual dose for young, healthy individuals is 2000 IU/day, thus the doses for our patients are higher due to the illness, as well as factors such as age and weight. Therefore, the dose increases with a higher BMI due to the increased amount of fat content in the body.

We have added some references and made the corrections in the 4.3 Intervention section.

“Supplementation lasted for 12 weeks for all groups. Supplementation of vitamin D3 was based on the participants' Body Mass Index (BMI). The BMI was measured by using the TANITA scale. The doses were set based on the literature. The recommended dose for healthy, young adults is 1000-2000 International Units (IU)/day [51]. For obese and overweight individuals, the doses are higher and range between 2500-4800 IU/day [52–54]. Taking these recommendations into consideration, along with the fact that serum concentration in PD patients is insufficient and lower than in healthy age-matched controls the following doses were prescribed for BMI under 25 – 4000 International Units (IU)/day, for BMI between 25 and 30 – 5000 IU/day, and for BMI over 30 - 6000 IU/day. The placebo group received a matching placebo treatment. Vitamin D3, as well as the placebo, were in the same bottles with no labels on them. Patients were prompted to do 3500 steps per day in the first week of the research and finish the research by making 7500 per day.”

Major remarks:

- The study itself is meticulously planned and has been registered as a clinical trial, which increases its credibility. Figures 5 amd 6 let the Readers follow the trial in a step-by-step manner. However, there are several potentially limiting factors which should be mentioned in the discussion section. Firstly, the supplementation with VitD varied depending on BMI - a factor which was higly heterogenic in the population sample (as seen in Table 1). Given that this factor affected the dosage of VitD, the population sample should be significantly greater than the one featured in this study. 'BMI-dose' should be an effect which interacts with 'time' in the repeated measured ANOVA. Moreover, as the population sample is unbalanced in terms of sex, the 'time*sex' interaction could not, likewise, be investigated. This laborous trial could have been more informative than just analyzing the time-wise variability in vitamin D3 and hsCRP concentrations. More timepoints, proper study design and thorough planning of the composition of the population sample would possibly allow drawing more interesting conclusions. As there is a lack of control group for the 'PD' effect (patients with no PD of the comparable population sample composition), it could not be answered whether the dynamics of the changes of some studied parameters (such as the concentration of VitD metabolites) is different in PD vs. no PD upon supplementation.

- The 'Statistical Analysis' section leaves as impression that the assumptions of the used methods were not thoroughly checked or accounted for (in case of their lack). RM-ANOVA assumes sphericity, which is often not found in empirical data. Therefore, corrections are performed to account for the lack of sphericity. I would advise performing the Greenhouse-Geisser or Huynh-Feldt corrections, being: more or less conservative, respectively). Since the Authors knew a priori that they wanted to check the time-wise differences, the utilization of contrast analysis would be more suitable than the Tukey's HSD post-hoc.

Thank you for the comments and suggestions.

The explanation related to BMI and vitamin D3 treatment is above. Based on your suggestions some changes related to vitamin D, BMI, and population have been incorporated into the manuscript.

“The doses were set based on the literature. The recommended dose for healthy, young adults is 1000-2000 International Units (IU)/day [51]. For obese and overweight individuals, the doses are higher and range between 2500-4800 IU/day [52–54]. Taking these recommendations into consideration, along with the fact that serum concentration in PD patients is insufficient and lower than in healthy age-matched controls the following doses were prescribed for BMI under 25 – 4000 International Units (IU)/day, for BMI between 25 and 30 – 5000 IU/day, and for BMI over 30 - 6000 IU/day”.

We agree with your comments. However, the population of PD patients with implanted DBS is limited. The study was performed in cooperation with the referral movement disease center, performing 20 DBS implantations in PD patients per year and taking care of about 200 PD patients with neurostimulators. 50 patients representing 25% of the local population were enrolled and 29 patients accomplished the study, that was partially performed during the SARS-COVID-2 pandemics. Regarding the vulnerability of these elderly patients to SARS-COVID-2 it was not ethical to continue enrollment after the onset of the pandemics. Still the group of 29 patients is the largest studied and results not only add to the knowledge on the role of VitD in PD but also translate to the potentially important clinical benefit – reduction of risk of fractures.

Based on those, we have run the stat analysis again and appropriate changes we have incorporated to the manuscript.

As suggested, we performed statistical analyses step by step, including checks for sphericity, Mauchly's corrections, and contrast analysis. For instance, the 6MW test results are attached below. I apologize but the description is written in the Polish language (the Polish version of Statistica software).

Despite that, we used Tukey's HSD (unequal N) post-hoc test because the power of this test is stronger.

We have changed the description as follows:

“For the analysis, we used the statistic programme Statistica 13, StatSoft Inc., Tulsa, OK, USA. We took under consideration only full data, from patients who completed all interventions (n=29). Data were previously tested for normality using Shapiro-Wilk W-test. The descriptive statistics for both background information and examination of the trends in the analysed parameters with mean values of 95% confidence interval were used. The ANOVA test for repeated measures was used for the statistical analysis. Spearman rank correlation was calculated. The statistical significance was set at p<0.05. To establish statistical significance we applied an analysis of variance (ANOVA) with Tukey’s post-hoc test.”

- I could not see in any of the plot descriptions what the whiskers mark. Are they refering to 95% CI of the mean, SD or SE? Time-related analysis should show SD and/or SE.

Thank you for the comments and suggestions. Based on those, we have run the stat analysis again and appropriate changes we have incorporated to the manuscript.  We have also changed the plot descriptions and explained the whiskers (refer to standard error; SE).

“Figure 1. The effect of supplementation on vitamin D metabolites. (A) Changes in serum concentrations of 25(OH)D3. (B) Changes in serum concentrations of 25(OH)D2. (C) Changes in serum concentration of 24,25(OH)2D3. (D) Changes in serum concentrations of epi-25(OH)D3. *p<0.0006, **p<0.005, ***p<0.05, T0 – before the supplementation, T2 – after 12 weeks of supplementation, PL – placebo group, VitD – vitamin D3 group, whiskers refer to standard error (SE).”

“Figure 3. Effect of supplementation on functional tests. A. Changes in time duration in TUG. B. Changes in the distance in 6 min walk test. C. Changes in time duration in 10 m walk test. *p<0.005, **p<0.05, TUG – test Up&Go, T0 – before the supplementation, T1 – after 6 weeks of supplementation, T2 – after 12 weeks of supplementation, PL – placebo group, VitD – vitamin D3 group, whiskers refer to SE.”

“Figure 4. Effect of inflammation on general inflammation status. hs-CRP – high sensitivity C-reactive protein, T0 – before the supplementation, T2 – after 12 weeks of supplementation, PL – placebo group, VitD – vitamin D3 group, whiskers refer to SE.”

- Line 389: The Pearson's r does not describe the rank correlation. It is based on residuals from the observed values vs. mean values.

Thank you for this comment, we have already changed the test that we used for correlation. So the sentence now is:

“Spearman rank correlation was calculated.”

Of course also the Figure 2 is changed.

“Figure 2. Relationships between (A) 25(OH)D3 and 24,25(OH)2D3; (B) 25(OH)D3 and epi-25(OH)D3. (A) Spearman r = 0.9, p<0.0001, 95% confidence interval: 0.83 to 0.94; (B) Spearman r = 0.92, p<0.0001, 95% confidence interval: 0.86 to 0.95.”

With my remarks in mind, although I believe that the study could be published in an indexed journal, I would not recommend publishing it in IJMS since the study does not investigate covariates and features results based on a small-size, heterogenic population sample.

I understand the concerns about the small sample size; however, there are several reasons why the sample size ended up being small.

Firstly, the main reason is that the Hospital of Nicholas Copernicus, specifically the Neurosurgery Unit, performs only around 20-22 deep brain stimulation (DBS) procedures annually under normal conditions. Due to the impact of the SARS-COVID-2 pandemic, the number of procedures during the study period was even lower. Moreover, this hospital is the only one within a 200 km area, limiting the pool of potential patients.

Secondly, the study included Parkinson's patients who had been qualified for deep brain stimulation by a neurologist working in another hospital. I would like to mention that sometimes, patients have to wait for approximately two years to undergo the DBS procedure. The limitations on the number of available DBS treatments at the Hospital of Nicholas Copernicus were beyond our control, which contributed to the heterogeneity of the sample.

Thirdly, recruitment for the study took place between November 2019 and February 2022. Additionally, the intervention was conducted during the fall and winter seasons to avoid exposing patients to sunlight during the spring and summer seasons.

Lastly, as mentioned earlier, some patients dropped out of the study due to various reasons, as described and detailed in the Methods and Materials section.

I hope that this explained why there is a small sample size. On the other hand in our opinion, we believe that the number of Parkinson’s patients who underwent DBS treatment was enough in that very strange situation.

Sincerely,

Corresponding Author: Jan J. Kaczor, PhD

Department of Animal and Human Physiology,

University of Gdansk.

Reviewer 2 Report

The idea is original and might be of interest, but I have big concerns:

It does not seem clear what the motor status of the patients is before and after the intervention, rather than the 3 functional tests, but in PD the most valid scores are UPDRS and HyY. There is also no reference to the duration of the disease in these patients, this is an important indicator of severity, beyond the time since the DBS.

The response to DBS prior to the intervention is not specified, nor are the parameters used.

The sample seems small to infer significant results

There are also some statements that are not accurate:

Line 17: "To control motor symptoms when drug treatments cause excessive side effects, deep brain stimulation 18 (DBS) is used." DBS is indicated to control motor fluctuations, not medication side effects.

Line 37: "PD in intractable"... PD is treatable but it is not possible to cure it

Line 45 - Again, it is mandatory to review the indication for DBS, there is no clear statement there

Line 65: paragraph. There are other well-known markers of peripheral inflammation in PD, such as the neutrophil to lymphocyte ratio (NLR)

None

Author Response

Reviewer#2

Dear Reviewer,

The authors of the manuscript would like to thank you for all your comments and suggestions concerning improvements of this manuscript. We followed them to revise the manuscript or tried to explain any ambiguities in the point-by-point response.

I hope that all changes, which have been made would be satisfactory to Reviewer#1.

The idea is original and might be of interest, but I have big concerns:

It does not seem clear what the motor status of the patients is before and after the intervention, rather than the 3 functional tests, but in PD the most valid scores are UPDRS and HyY

Thank you for this suggestion. The information regarding H&Y scores in the studied patients based on medical records has been added. The additional line has been included in Table 1 and the description in the 2.1 Demographical characteristics have been corrected.

VitD group (n=13)

PL group (n=16)

Age (years)

63 ± 9

66 ± 6

Sex

6 M, 7 W

13 M, 3 W

Height (cm)

169 ± 12

174 ± 8

Body mass (kg)

78 ± 11

80 ± 20

 DBS implantation

3-5 years ago

3-5 years ago

H&Y

2,5

2,5

Duration of the disease

8 – 13 years

8 – 13 years

BMI

≤ 25 → 3

25 – 30 → 8

≥ 30 → 2

≤ 25 → 6

25 – 30 → 8

≥ 30 → 2

“The time from DBS implantation varied from three to five years in both groups. In the off-medication condition mean motor improvement after DBS stimulation, as assessed with UPDRS part 3, was 45% (range 35-60%). The mean L-dopa-equivalent medication dosage reduction was 35%. Stimulation parameters were: monopolar stimulation in all patients, frequency 130 Hz, pulse duration 60 us, current 1.8 – 3.8 mA (in 20 patients with constant-current stimulators), and voltage 1.9 – 4.2 V (in 9 patients with constant-voltage stimulators). Hoehn and Yahr (H&Y) scores in both groups were 2.5. Duration of the disease varies from 8 to 13 years in the VitD and PL groups. In the VitD group, there were 3 patients with normal weight, 8 patients with flesh, and 2 with obesity. In the PL group, there were 6 patients with normal weight, 8 patients with overweight, and 2 patients with obesity. The characteristics of patients are shown in Table 1.”

As our study focused on the physical performance (influenced by vitamin D) rather than neurological motor symptoms (influenced by antiparkinsonian medication and DBS), functional tests have been used to assess motor improvement after VitD supplementation. UPDRS, as more dependent on DBS and medication might not reflect adequately metabolic changes produced by vitamin D. H&Y score reflects the stage of the disease and is not expected to change after supplementation of VitD.

There is also no reference to the duration of the disease in these patients, this is an important indicator of severity, beyond the time since the DBS.

Thank you for this comment. This information has been added in the Table 1 and in the 2.1 Demographical characteristics section. 

VitD group (n=13)

PL group (n=16)

Age (years)

63 ± 9

66 ± 6

Sex

6 M, 7 W

13 M, 3 W

Height (cm)

169 ± 12

174 ± 8

Body mass (kg)

78 ± 11

80 ± 20

 DBS implantation

3-5 years ago

3-5 years ago

H&Y

2,5

2,5

Duration of the disease

8 – 13 years

8 – 13 years

BMI

≤ 25 → 3

25 – 30 → 8

≥ 30 → 2

≤ 25 → 6

25 – 30 → 8

≥ 30 → 2

“The time from DBS implantation varied from three to five years in both groups. In the off-medication condition mean motor improvement after DBS stimulation, as assessed with UPDRS part 3, was 45% (range 35-60%). The mean levodopa-equivalent medication dosage reduction was 35%. Stimulation parameters were: monopolar stimulation in all patients, frequency 130 Hz, pulse duration 60 us, current 1.8 – 3.8 mA (in 20 patients with constant-current stimulators), and voltage 1.9 – 4.2 V (in 9 patients with constant-voltage stimulators). Hoehn and Yahr (H&Y) scores in both groups were 2.5. Duration of the disease varies from 8 to 13 years in the VitD and PL groups. In the VitD group, there were 3 patients with normal weight, 8 patients with flesh, and 2 with obesity. In the PL group, there were 6 patients with normal weight, 8 patients with overweight, and 2 patients with obesity. The characteristics of patients are shown in Table 1.”

The response to DBS prior to the intervention is not specified, nor are the parameters used.

Thank you for this reflection, we have added this information in the 2.1 Demographical characteristics.

“The time from DBS implantation varied from three to five years in both groups. In the off-medication condition mean motor improvement after DBS stimulation, as assessed with UPDRS part 3, was 45% (range 35-60%). The mean levodopa-equivalent medication dosage reduction was 35%. Stimulation parameters were: monopolar stimulation in all patients, frequency 130 Hz, pulse duration 60 us, current 1.8 – 3.8 mA (in 20 patients with constant-current stimulators), and voltage 1.9 – 4.2 V (in 9 patients with constant-voltage stimulators). Hoehn and Yahr (H&Y) scores in both groups were 2.5. Duration of the disease varies from 8 to 13 years in the VitD and PL groups. In the VitD group, there were 3 patients with normal weight, 8 patients with flesh, and 2 with obesity. In the PL group, there were 6 patients with normal weight, 8 patients with overweight, and 2 patients with obesity. The characteristics of patients are shown in Table 1.”

The sample seems small to infer significant results

The population of PD patients with implanted DBS is limited. The study was performed in cooperation with the referral movement disease center, performing 20 DBS implantations in PD patients per year and taking care of about 200 PD patients with neurostimulators. 50 patients representing 25% of the local population were enrolled and 29 patients accomplished the study, that was partially performed during the SARS-COVID-2 pandemic. Regarding the vulnerability of these elderly patients to SARS-COVID-2 it was not ethical to continue enrollment after the onset of the pandemic. Still, the group of 29 patients is the largest studied and results not only add to the knowledge on the role of VitD in PD but also translate to the potentially important clinical benefit – reduction of risk of fractures.

There are also some statements that are not accurate:

Line 17: "To control motor symptoms when drug treatments cause excessive side effects, deep brain stimulation 18 (DBS) is used." DBS is indicated to control motor fluctuations, not medication side effects.

This statement has been corrected after your suggestion. Thank you.

“To manage motor symptoms not controlled adequately with medication, deep brain stimulation (DBS) is used.”

Line 37: "PD in intractable"... PD is treatable but it is not possible to cure it,

This statement has been corrected after your remark. Thank you.

“Currently, PD is incurable, and the main aim is to palliate symptoms, ease suffering and slow down the development of the disease, in order to help the patients to cope better with their daily living activities.”

Line 45 - Again, it is mandatory to review the indication for DBS, there is no clear statement there,

This statement has been corrected after your suggestions. Thank you.

“When pharmacological treatment ceases to control symptoms adequately, deep brain stimulation (DBS) is used to mitigate motor symptoms and decrease drug dosage.”

Line 65: paragraph. There are other well-known markers of peripheral inflammation in PD, such as the neutrophil to lymphocyte ratio (NLR),

Thank you for your remark. The general inflammation status was the thing that we wanted to check that is why we decided to measure the hs-CRP concentration.

Sincerely,

Corresponding Author: Jan J. Kaczor, PhD

Department of Animal and Human Physiology,

University of Gdansk.
